# Clinical Outcomes in AYAs (Adolescents and Young Adults) Treated with Proton Therapy for Uveal Melanoma: A Comparative Matching Study with Elder Adults

**DOI:** 10.3390/cancers15184652

**Published:** 2023-09-20

**Authors:** Alessia Pica, Damien C. Weber, Claude Schweizer, Aziz Chaouch, Leonidas Zografos, Ann Schalenbourg

**Affiliations:** 1Center for Proton Therapy, Paul Scherrer Institute, ETH Domain, 5232 Villigen West, Switzerland; damien.weber@psi.ch; 2Department of Radiation Oncology, Inselspital, Bern University Hospital, University of Bern, 3012 Bern, Switzerland; 3Department of Radiation Oncology, University Hospital of Zurich, 8091 Zurich, Switzerland; 4Department of Ophthalmology, University of Lausanne, Jules-Gonin Eye Hospital, FAA (Fondation Asile des Aveugles), 1004 Lausanne, Switzerland; claude.schweizer@fa2.ch (C.S.); l.zografos@oncol.ch (L.Z.); ann.schalenbourg@fa2.ch (A.S.); 5Department of Epidemiology and Health Systems, Center for Primary Care and Public Health (Unisanté), University of Lausanne, 1011 Lausanne, Switzerland; aziz.chaouch@unisante.ch

**Keywords:** eye tumor, uveal melanoma, proton therapy, AYA, TYA, teenagers, adolescents, young adults, propensity score matching, PSM, survival, metastasis incidence, local tumor control

## Abstract

**Simple Summary:**

This study assesses for the first time the clinical outcomes in a group of adolescent and young adult (AYA: 15–39 years old) uveal melanoma (UM) patients (n = 270) treated with proton therapy, comparing them with matched elder adults (>40 years). Ocular outcomes, local tumor control, metastasis incidence, and overall survival were similar in both groups, in contrast with the better survival rates in previous, pediatric UM series. Moreover, when compared with the general population, the AYAs had reduced relative survival, while the elder adults had better relative survival.

**Abstract:**

Objective: The aim of this study was to compare the clinical outcomes of adolescents and young adults (AYAs) with those of elder adult patients treated with proton therapy (PT) for uveal melanoma (UM). Material and Methods: A retrospective, comparative study was conducted in UM patients who underwent PT at the Ocular Oncology Unit of the Jules-Gonin Eye Hospital (University of Lausanne, Lausanne, Switzerland) and the Paul Scherrer Institute (PSI); (Villigen, Switzerland) between January 1997 and December 2007. Propensity score matching (PSM) was used to select for each AYA (between 15–39 years old) an elder adult patient (≥40 years) with similar characteristics. We assessed ocular follow-up, local tumor control, metastasis incidence, and overall and relative survival (OS and RS). Non-terminal outcomes were then compared between the two groups using competing risk survival analysis. Results: Out of a total of 2261 consecutive UM patients, after excluding 4 children (<15 years) and 6 patients who were metastatic at presentation, we identified 272 AYA patients and matched 270 of them with 270 elder adult patients. Before PSM, the AYA patients had a higher incidence of primary iris melanoma (4.0% vs. 1.4%; *p* = 0.005), while the elder patients were more likely to have other neoplastic diseases at presentation (9% vs. 3.7%; *p* = 0.004). Ocular outcomes and local tumor control were similar in both groups. Cumulative metastasis incidence for the AYA and elder adult groups was 13% and 7.9% at 5 years and 19.7% and 12.7% at 10 years, respectively, which was not significantly different between the groups (*p* = 0.214). The OS was similar in the two groups (*p* = 0.602), with estimates in the AYA and elder adult groups of 95.5% and 96.6% at 5 years and 94.6% and 91.4% at 10 years, respectively. However, the relative survival (RS) estimation was worse in the AYA group than the elder group (*p* = 0.036). Conclusion: While AYAs treated with PT for UM have similar ocular outcomes and present the same metastasis incidence and OS as elder adults, their RS is worse than that in elder adults, when compared with the population in general.

## 1. Introduction

Recently, an increased awareness has risen with regard to the specific needs of adolescents and young adults (AYAs) suffering from cancer. While cancer survival has significantly increased in children and elder adults over the last few decades, numerous studies have suggested that AYAs, defined as individuals with an age of 15 to 39 years at the time of initial cancer diagnosis, do not share these improved outcomes [1].

Uveal melanoma (UM) is the most common primary ocular malignancy in adults, with an overall incidence of about 5–9 new cases per million per year in the US and Europe [2]. The incidence of this tumor is relatively rare in young people, with fewer than 1 in every 100 UM patients being younger than 21 years [3,4]. There are a few reports describing the clinical and histopathological features of UM in young patients [3,4,5]. Radiation therapy (RT) represents the standard care for the vast majority of UM. Proton therapy (PT) is the most widely used form of RT because of its intrinsic physical properties, whereby the deposition of most energy occurs at the target tissue with a reduction of the dose proximal and distal to the tumor. PT is associated with high local tumor control rates and has been used for several decades in our center as the preferred RT modality of for UM [6,7]. Previous studies have found that ophthalmic outcomes of UM in children and AYAs do not differ from those in elder adults [3,8], while the metastatic-free survival of children with UM until the age of 16 seems to be much better. A survey by the European Ophthalmic Oncology Group compared the prognosis of UM patients less than 18 years old with those aged between 18 and 24 years, finding a 10-year melanoma-related survival rate of 92% and 80%, respectively [5].

In 2017, a joint Cancer in AYA working group was established, by the ESMO (European Society for Medical Oncology) and SIOPE (European Society for Pediatric Oncology) [9]. Inspired by their recommendations, we decided, to our knowledge for the first time, to focus on UM in this intermediate age group. The aim of this study is to compare the long-term ocular follow-up, local tumor control, and metastatic and survival rates for a group of AYAs and a matched control group of elder adult patients following proton therapy for UM.

## 2. Materials and Methods

### 2.1. Study Design: Patients

This retrospective, comparative study retrieved the clinical data from 2261 consecutive patients who had been treated with PT for primary UM at the Ocular Oncology Unit of the Jules-Gonin Eye Hospital (University of Lausanne, Lausanne, Switzerland) and the Paul Scherrer Institute (PSI) (Villigen, Switzerland) between January 1997 and December 2007. Patients who, at the time of their UM diagnosis, were metastatic (n = 6 cases, all elder than 15 years), or who were younger than 15 years old (n = 4 cases), were excluded from this analysis, leaving 2251 cases.

These 2251 patients were then divided into the AYA group and the elder adult group according to their age on the day of tantalum clip surgery, i.e., between 15 and 39 years old and equal to or elder than 40 years old, resulting in 272 cases in the former and 1979 cases in the latter group, respectively.

Patients’ characteristics included age, sex, and country of residence, classified into Swiss residents, residents of a bordering country, and residents of a country not sharing a border with Switzerland (this variable was included because, at Jules-Gonin/PSI, many international patients are treated, and the travel distance to Lausanne, as well as differences in the healthcare systems, may impact adherence to follow-up recommendations) [6]. Certain other data, with regard to systemic and ocular history, including vision, were also recorded (Table 1). Tumor characteristics included whether the UM presumably originated within the iris (iris melanomas having a different TNM classification). To classify the tumor, we used the 8th edition of the TNM classification by the American Joint Committee on Cancer Staging of UM [10]. Tumor location was additionally categorized according to its anterior margin extending into the iris, ciliary body, anterior choroid, or posterior choroid. Tumor dimensions were defined by its largest and smallest basal diameter, measured during tantalum clip surgery, and its thickness on B-scan ultrasonography. For diffuse iris melanomas, the largest and smallest tumor diameters were set to 12 mm for this analysis. Several other ophthalmological observations specifying tumor presentation within the affected eye were also included (Table 1).

This study complied with the Declaration of Helsinki and was provided a waiver of consent by the competent review board (ethical committee of the Canton of Vaud, Switzerland; authorization #2016–01861).

### 2.2. Treatment and Follow-Up

Clinical examination at baseline, during tantalum clip surgery, and at follow-up visits took place at the Ocular Oncology Unit of the Jules-Gonin Eye Hospital. PT was performed at the Paul Scherrer Institute, with the OPTIS system, delivering 60 Gy (RBE) in four fractions, on four consecutive days [7]. A standard follow-up visit included evaluation of Snellen BCVA (best-corrected visual acuity), slit-lamp examination, intra-ocular pressure measurement, and dilated indirect ophthalmoscopy, along with color fundus photography and B-scan ultrasonography. Ophthalmic examinations were performed before tantalum clip surgery, 6 months after PT and then at least annually for 15 years. Metastatic screening, consisting principally of liver imaging (ultrasonography and/or computed tomography scan and/or magnetic resonance imaging) was performed before treatment, twice a year during the first five years and then once a year. For patients unable to return to Lausanne for follow-up visits, information was gathered by contacting their local physicians.

### 2.3. Propensity Score Matching (PSM) and Statistical Analysis

AYA and elder adult patients were compared according to several covariates. Associations between group membership and categorical variables were evaluated using a chi-square test with Yates continuity correction or Fisher exact test, while for those involving continuous variables were evaluated using a two-sample *t*-test or a Mann–Whitney test depending on the distribution (Table 1). Paired tests were used for secondary outcomes (Table 2).

Propensity score matching (PSM) [11] was used to select elder adult patients with characteristics similar to those of the AYAs, taking into account certain patient variables (sex, country of residence, history of familial UM, and presence of other known tumors), year of treatment, and several tumor characteristics (presumed iris origin, location of anterior margin, largest and smallest basal diameter, thickness, extra-scleral extension, and TNM classification). The overlap of propensity score distributions in the two groups was assessed using density plots. Covariate balance after matching was verified by calculating standardized mean differences (SMD), assuming that an SMD <0.1 is indicative of suitable covariate balance [12].

The cumulative incidence function (CIF) [13] was estimated for non-terminal outcomes (i.e., metastasis, local recurrence, actinic maculopathy, and neuropathy) in the two groups while accounting for competing risks. For instance, death was considered as a competing risk for all non-terminal outcomes as dead patients are no longer at risk of experiencing these outcomes. Similarly, enucleation was considered as a competing risk for a later local recurrence, actinic maculopathy, or neuropathy within the enucleated eye. In order to account for dependences introduced by matched pairs, a formal comparison between the CIFs in the two groups was carried out using an extension of the Fine–Gray model for clustered data [14], as implemented in the *R* package *crrSC* [15].

Overall survival (OS) in the two groups was estimated using Kaplan–Meier curves. The difference between these curves was tested using a log-rank test adapted to clustered data [16]. However, note that the two groups essentially differ by age and that age is a strong determinant of OS. In consequence, comparing the OS in the two groups on the sole basis of Kaplan–Meier curves may not properly reflect cancer-related survival. Therefore, we estimated the relative survival (RS) in each group by comparing the observed survival (i.e., OS on our Kaplan–Meier curves) with the expected survival in the general population, all the while correcting for age, sex, the country of residence, and year of treatment. Information on the expected survival was derived from national lifetables downloaded from the Human Mortality Database [17]. Swiss residents’ survival was compared to lifetables of Switzerland, whereas residents of bordering countries were compared to lifetables of Italy and residents of non-bordering countries were compared to lifetables of Greece, as an approximation, because the majority of residents of a bordering country were Italian residents, and the majority of residents of a non-bordering country were residents of Greece. RS in each group was estimated for each year of follow-up using the Ederer II estimator (see [18] for implementation details). The difference between RS curves in the two groups was tested by calculating the sum of absolute differences between yearly estimates of RS in the two groups (i.e., test statistic), and then comparing this sum with the distribution of the test statistic under the null hypothesis of no difference in RS. The distribution of the test statistic under the null hypothesis was constructed using a permutation test where group membership was randomly switched within each pair of matched patients. A total of 10,000 permutation samples were constructed in this way. The *p*-value of the test was estimated by calculating the proportion of permuted samples leading to a test statistic that was equal to or larger than that obtained in the matched sample. Results were deemed statistically significant for *p* < 0.050. All statistical analyses were carried out in R version 4.1.2 [19].

## 3. Results

### 3.1. Matching the AYAs with Elder Adults, Using PSM

The patients’ baseline ocular, treatment, and tumor characteristics before PSM are detailed in Table 1. The median patient age at the start of treatment in the AYA (n = 272) and elder adult (n = 1979) groups was 34 years (IQR: [29–37]) and 61 years (IQR: [53–69]), respectively. A total of 6 (2.2%) patients in the AYA group and 34 (1.7%) patients in the elder adult group had a family history of uveal melanoma (*p* = 0.621), which was not statistically significant. However, the latter were more likely to have other neoplastic diseases at presentation (9.0% vs. 3.7%; *p* = 0.004). They also had more chances that their tumor was discovered before they presented with symptoms *(p* = 0.008). The most significant difference in tumor characteristics was that, compared with their elders, AYAs had a higher incidence of primary iris melanoma (4.0% vs. 1.4%; *p* = 0.005). In addition, AYAs were more likely to present a tumor invasion of the macula: 17.6% vs. 12.8% (*p* = 0.034). On the other hand, the probability of the tumor disrupting (that is, invading and growing through) the retina (Knapp–Rønne-type UM) was higher in the elder adults (12.1% vs. 6.6%; *p* = 0.010).

After PSM was performed (Table 1), taking into account sex, country of residence, a positive history of familial uveal melanoma, the presence of other known tumors, year of proton therapy, and certain tumor characteristics (presumed iris origin, location of anterior tumor margin, largest and smallest tumor diameter, thickness, the presence of an extra-scleral extension, and TNM classification), the AYAs (270 patients) and elder adults (270 patients) groups became balanced, with standardized mean differences (SMD) <0.1 achieved on all matched variables.

### 3.2. Comparing Clinical Outcomes between the AYAs and Elder Adults

Table 2 summarizes clinical follow-up data, comparing the AYAs and matched elder adults with regard to both systemic and ocular outcomes.

#### 3.2.1. Ocular Follow-Up and Local Tumor Control

Apart from the probability of developing cataract, a definitely age-related eye condition, no major differences could be observed during the ocular follow-up between both groups.

Nine AYAs (3.3%) and five elder adults (1.9%) presented a local tumor recurrence within a mean follow-up time of 8.9 ± 7.7 and 6.5 ± 2.7 years, respectively. The difference between the groups was not significant (*p* = 0.96). These recurrences were treated with a second proton beam irradiation in seven (0.3%) patients (four AYAs and three elder adults) and with enucleation in five (0.2%) patients (three AYAs and two adults, including one after a second circumscribed PT for recurrent ciliary body melanoma). For three patients, no data on the type of salvage treatment were available.

#### 3.2.2. Metastasis Incidence

As already mentioned, there was no evidence of metastasis in the patients of this study at the time of their UM diagnosis. The mean (SD) time before metastasis discovery in the AYA and elder adult groups was 3.5 (2.5) and 3.7 (2.9) years, respectively. The cumulative incidence of metastasis for the AYAs was 13% at 5 years (95% CI = 9.0–18.7) and 19.7% at 10 years (95% CI = 14.1–27.6), while it was 7.9% at 5 years (95% CI = 4.8–13.2) and 12.7% at 10 years (95% CI = 7.9–20.4) for the elder adults. These differences were not significant (*p* = 0.214) between the two groups. In Figure 1, the cumulative incidence for metastasis in both groups is shown.

#### 3.2.3. Overall Survival (OS) and Relative Survival (RS)

The median follow-up time for the whole cohort and for the surviving patients was 4.5 years (range, 0, 20.8) and 4.8 years (range, 0, 20.8), respectively. Ten deaths were observed in both the AYA and elder adult groups. OS did not differ between the two groups (*p* = 0.602) (Figure 2), with an estimated 5-year OS of 95.5% (95% CI = 92.4–98.6) and 96.6% (95% CI = 94.0–99.3) in AYAs and elder adults, respectively, while the 10-year OS was 94.6% and 91.4%.

We estimated the relative survival (RS) in each group by comparing the observed survival (i.e., on our Kaplan–Meier survival curve) with the expected survival in the general population.

The estimated 5-year RS in the AYA and elder adult groups was 0.96 (95% CI = 0.93–0.99) and 1.06 (95% CI = 1.00–1.06), respectively. Relative survival estimates were better for the elder adult patient group (*p* = 0.036) (Figure 3).

## 4. Discussion

This retrospective study compares the clinical outcomes of 270 AYAs with those of 270 propensity score-matched elder adults with UM, who were treated with proton therapy between January 1997 and December 2007.

### 4.1. Demographic, Patient, and Tumor Characteristics

When compared with children or elder adults, AYA cancer patients are recognized as a distinct population within the oncology community due to the unique challenges they face during and after their disease, including limited autonomy, in-progress education, and family planning. It is also noteworthy [20] that AYA patients have generally an inferior survivorship when compared with the general population and children cancer survivors, the reasons for the latter being the lack of standardized care delivered by non-pediatric oncologists and/or the low inclusion in treatment protocols, to name only a few reasons [21].

Uveal melanoma remains a rare disease with an annual incidence of 5–9 per million in North America and Europe and a low incidence before 45 years of age. Between 1973 and 1997, the calculated age-specific incidence of UM in the USA, per million population, was 0for 0–4 years, 0.2 for 10–14 years, and 0.4 for 20–24 years. After these age cut-offs, the incidence continues to increase with age [22]. In our study, only 272 (12%) AYAs out of 2261 UM patients were treated during the 11-year study interval. In rare instances, UM occurs in the presence of predisposing factors (e.g., oculo(dermal) melanocytosis, dysplastic nevus syndrome, neurofibromatosis type 1 and germline mutations in the BRCA1-associated protein 1 (BAP1) gene [23]. Very rare instances of familial occurrence of melanoma support the concept of some genetic predisposition [24]. Interestingly, we did not observe an increase of familial UM in the AYAs (2.2%) when compared with the adult cohort (1.7%; *p* = 0.744).

In the present study, the AYA group had higher incidence of primary iris melanoma, which is in accordance with several other studies showing that patients with iris melanoma are about a decade younger at presentation than those with posterior (i.e., ciliary body and choroidal) uveal melanoma. However, we did not find a gender predilection for the development of iris melanoma, which is in line with other series [25].

### 4.2. Ocular Follow-Up and Local Tumor Control

Our study confirmed the findings of previous reports [3,8] that the ocular outcomes are not significantly influenced by the age of UM patients.

Local recurrences were diagnosed in 14 out of 540 eyes (2.6%). We observed that the local failure rate in AYAs was approximately double the rate in elder adults, but this difference did not reach statistical significance. We obtained results comparable with our previous outcomes published in 2001 by Egger et al. [7]. In our report on 43 juvenile (younger than 21 years) UM patients, where each juvenile patient had been matched with three adult control patients, the rates of local tumor recurrence were also similar in the two groups [3]. In 2016, a European retrospective multicenter observational study comparing 114 children (less than 18 years) and 185 young adults (less than 25 years), local UM recurrence was diagnosed in 7 children (6%) and 9 young adults (5%), which was not statistically different from the elder patients’ rate [5].

### 4.3. Metastasis Incidence

The role of increasing age as a predictor of UM metastasis is still debatable. In Table 3, we summarize the conclusions of the more recent publications on this subject. Younger patient age at the time of diagnosis and treatment has been identified by some as a favorable prognostic factor [3,5,26,27,28]. In a matched retrospective cohort study (122 patients in each group) on children and adolescents (<20 years) versus mid-adults (21–60 years) versus elder adults (>60 years), the metastatic rate, after adjusting for tumor diameter, was found to be lower in children compared to mid-adults (*p* = 0.042, HR 3.00) and elder adults (*p* = 0.007, HR 4.20). Using univariate and multivariate analysis the authors claimed that increasing age was the main predictive factor for tumor metastasis in young patients [26]. Similar to these findings, Petrovic et al. concluded that the metastatic rate at 10 years was significantly lower in juvenile UM patients than in adult controls (11% vs. 34%, *p* < 0.01) [3]. They also observed, after splitting the juvenile group into children (<16 years) and young adults (16–20 years), that none of the patients diagnosed before the age of 16 years developed metastatic disease, as confirmed by Al-Jamal et al. in 15 children younger than 10 years, during their European Ophthalmic Oncology Group review on 299 UM patients less than 25 years old [5]. Only Fry et al. in a report on 18 patients, of whom 14 were elder than 15 years, questioned this ‘favorable’ prognosis, albeit without having the numbers to perform a multivariate analysis [8].

In the current AYA study, increasing age was not correlated with a higher rate of metastasis; the difference with the elder adult group was not significant. This observation favors the theory of a ‘threshold age’ rather than that of a linear age effect on metastatic risk. While previous studies found that prepubescent UM patients seem to be relatively protected from developing metastases, we selected for our study the distinct AYA population aged 15–39 years, recently identified for presenting challenging tumors with clinical and biological differences, which are potentially more aggressive and more refractory to treatment than those in the elder adults. Noteworthy is the fact that few patients had a follow-up of more than 10 years, while Lane et al. observed, in a cohort of 3088 UM patients, that cumulative melanoma-related mortality rates continued to increase up to 23 years after treatment. Although annual rates decreased considerably (to <1%) 14 years after treatment, their data highlight the importance of a long-term follow-up [31].

### 4.4. Overall Survival (OS) and Relative Survival (RS)

As a mirror to the relatively lower metastatic risk, previous studies have also consistently shown that UM children have a more favorable survival compared to that of elder patients, while, in our AYAs study, the overall 5- and 10-year survival (OS) was similar to that of elder adults (Table 3).

In order to negate the effect of age on OS, we also computed the relative survival (RS). Although the 95% confidence intervals for the RS curves are fairly wide (Figure 3), we observe that the RS estimate among patients elder than 40 years generally sits above 1, while the RS estimate in the AYA group tends to gradually descend below 1 over time. This suggests that patients elder than 40 years in our cohort are slightly less at risk of dying while AYAs have a higher risk compared to the general population in the same age range.

One possible explanation is that we used all-cause mortality to calculate RS. First, it should be noted that deaths due either directly or indirectly to uveal melanoma only represent a fraction of all-cause mortality in the general population. Nevertheless, all-cause mortality predominantly affects the elderly while deaths in the young population (<40 years old) are rather seldom in the general population. Therefore, only a few deaths among AYA patients will drive the relative survival below 1 as the risk of death is very low for the corresponding age group in the general population. In contrast, all-cause mortality in the general population over 40 years increases substantially with age. Due to their condition, patients in our cohort have a tight medical follow-up with regular medical visits (including blood testing) and are routinely screened for metastasis. This allows the early detection of potentially existing comorbidities (e.g., cardiovascular diseases or other cancers), with the potential possibility to take preventive/corrective actions to mitigate the risk associated with these conditions. The general population does not, however, benefit from such level of medical attention, with many medical conditions remaining undiagnosed until later in life. Second, we speculate that cancer patients might be more inclined to adopt a healthier lifestyle compared to that of the general population, which may prevent the occurrence other health problems (e.g., cardiovascular diseases). In other words, while being obviously more at risk of dying from uveal melanoma or its consequences, we hypothesize that patients in our cohort are likely less at risk of dying from other causes than the general population. This may translate into a relative survival above 1 in the elder age group because it is mostly within this age group that mortality from other causes is relevant in the general population.

Limitations of this study include those linked to its retrospective nature. Because of the rarity of UM, specifically in younger patients, as well as the importance of a long follow-up with regard to metastasis-related survival, a randomized clinical trial is not feasible. By opting for PSM, we made every effort to avoid the effect of sources of bias and variability originating from the use of historical data or possible differences in adherence or attendance on the study outcomes. An important limitation, correlated to the retrospective nature of the study, is the report of the real number of deceased patients. Likely, some patients were lost to follow-up because they died, and we did not receive the information.

As we see it, implications of this study, apart from being the first to focus on UM outcomes in AYAs, could be linked to the confirmation of an apparent ‘threshold’ age for UM metastasis, rather than a metastatic risk gradually increasing with age. We do not know why, as age-specific molecular and genetic features are poorly understood. Ideally, all patients should undergo mutation profiling as part of their diagnostic work-up to identify tumor biology and prognostic biomarkers for metastatic disease. As this currently requires an intraocular tumor biopsy, such an approach is limited in the presence of small-sized tumors, as well as by the risks of sub-retinal or vitreous hemorrhage and tumor dissemination. More promising would be the implementation of less invasive strategies to determine the metastatic potential of UM patients in the absence of a tumor biopsy.

## 5. Conclusions

Our study demonstrates that AYA and elder adult UM patients treated with proton therapy encountered similar risks of local recurrence and probably distant metastasis. In contrast with the better survival outcomes in studies on pediatric UM, overall survival was similar in these both groups. Moreover, relative survival estimates were significantly worse in AYAs than in elder adult patients.

## Figures and Tables

**Figure 1 cancers-15-04652-f001:**
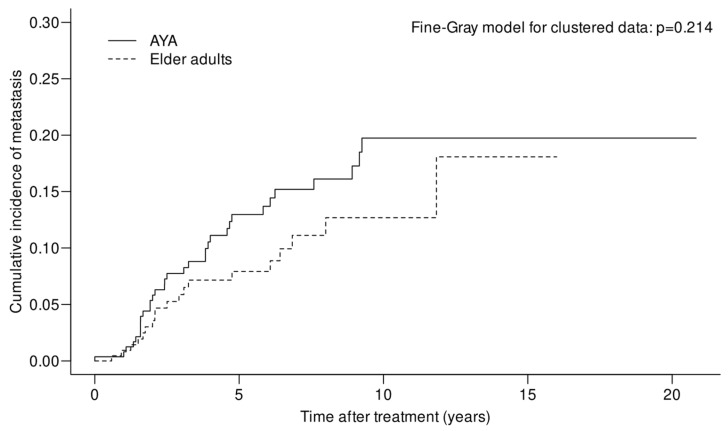
Cumulative incidence for metastasis in the AYA and matched elder adult groups.

**Figure 2 cancers-15-04652-f002:**
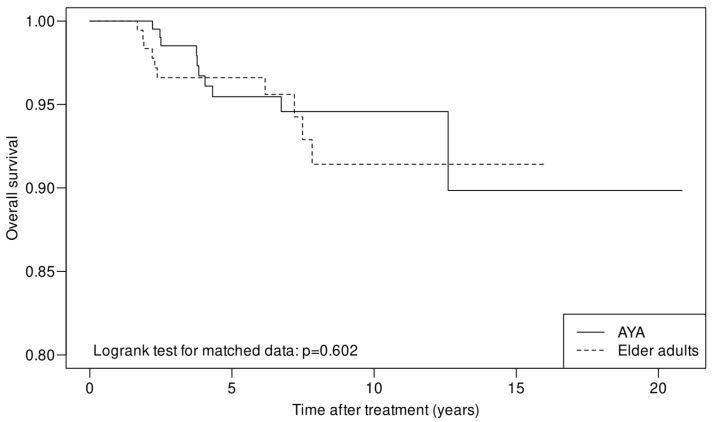
Patient overall survival in the AYA and matched elder adult groups.

**Figure 3 cancers-15-04652-f003:**
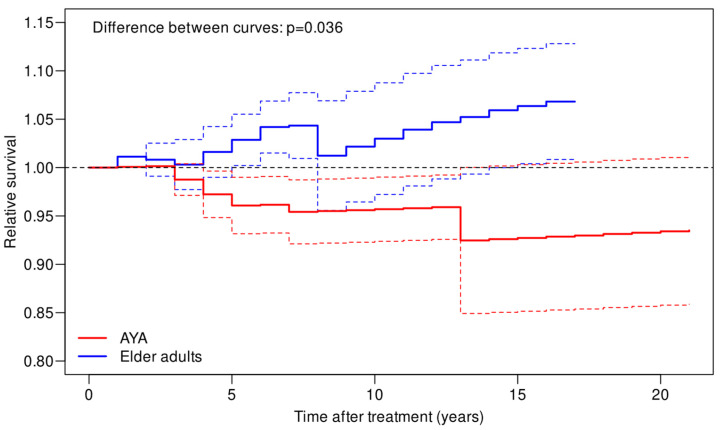
Relative survival of the AYA and elder adult UM patients with respect to the general population of the same age. Dashed lines delimit a 95% confidence interval for the relative survival. The difference between the two relative survival curves was tested using a permutation test. The dashed black line indicates the survival in the general population.

**Table 1 cancers-15-04652-t001:** Baseline patients, ocular, treatment, and tumor characteristics comparing AYAs and elder adults before and after matching.

	Before Matching			After Matching			
	AYAs	Elder Adults	*p*-Value	AYAs	Elder Adults	SMD	*p*-Value
**Patients, n**	**272**	**1979**		**270**	**270**		
Age in years,							
Mean ± SD	32.5 ± 5.4	61.1 ± 10.9		32.5 ± 5.4	60.5 ± 10.8		
Median [IQR]	34 [29–37]	61 [53–69]	**<0.001** ^a^	34 [29–37]	61 [52–69]		
Range	16–39	40–91		16–39	40–91		
Sex, n (%) *			0.056 ^b^			0.053	
Male	112 (41.2)	941 (47.5)		111 (41.1)	104 (38.5)		
Female	160 (58.8)	1038 (52.5)		159 (58.8)	166 (61.5)		
**Country of residence**, n (%) *			0.507 ^b^			0.009	
Switzerland	58 (21.3)	465 (23.5)		58 (21.5)	59 (21.9)		
Bordering country	140 (51.5)	1034 (52.2)		139 (51.5)	138 (51.1)		
Other	74 (27.2)	480 (24.3)		73 (27.0)	73 (27.0)		
**Systemic and ocular history**, n (%)							
Familial uveal melanoma *	6 (2.2)	34 (1.7)	0.621 ^c^	6 (2.2)	8 (3.0)	0.047	
Other known tumors *	10 (3.7)	179 (9.0)	**0.004** ^b^	10 (3.7)	8 (3.0)	0.041	
Ocular melanocytosis	7 (2.6)	47 (2.4)	1.000 ^b^	7 (2.6)	3 (1.1)		
Patient presented with symptoms	238 (87.5)	1596 (80.6)	**0.008** ^b^	236 (87.4)	224 (83.0)		0.15 ^e^
Documented tumor growth (i.e., treatment after an observation period)	53 (19.5)	317 (16.0)	0.174 ^b^	52	53		0.91 ^e^
**Eye laterality**							
Right eye, n (%)	136 (50.0)	984 (49.7)	0.983 ^b^	135 (50)	124 (45.9)		
**BCVA (Snellen)** in the affected eye							
≥0.1 (20/200), n (%)	251 (92.3)	1759 (88.9)	0.111 ^b^	249 (92.2)	242 (89.6)		0.30 ^e^
≥0.6 (20/33), n (%)	162 (59.6)	1120 (56.6)	0.389 ^b^	160 (59.3)	155 (57.4)		0.65 ^e^
**Proton therapy**							
Year of PT, median [IQR] *	2001 [1999–2004]	2002 [1999–2005]	**0.014** ^a^	2001 [1999–2004]	2001 [1999–2004]	0.035	
More than 3 months delay between diagnosis and treatment, n (%)	5 (1.8)	48 (2.4)	0.700 ^b^				
**Tumor location**							
Origin within the iris, n (%) *	11 (4.0)	28 (1.4)	**0.005** ^c^	9 (3.3)	11 (4.1)	0.051	
Anterior tumor margin within the *			**<0.001** ^b^			0.048	
iris, n (%)	33 (12.1)	117 (5.9)		31 (11.5)	33 (12.2)		
ciliary body, n (%)	66 (24.3)	518 (26.2)		66 (24.4)	67 (24.8)		
anterior choroid, n (%)	58 (21.3)	574 (29.0)		58 (21.5)	61 (22.6)		
posterior choroid, n (%)	115 (42.3)	770 (38.9)		115 (42.6)	109 (40.4)		
**Tumor dimensions** in mm, mean ± SD							
LTD *	16.2 ± 4.2	16.3 ± 4.2	0.641 ^d^	16.2 ± 4.2	16.4 ± 4.3	0.047	
STD *	13.9 ± 3.7	14.0 ± 3.8	0.762 ^d^	13.9 ± 3.7	13.9 ± 3.8	0.016	
Tumor thickness *	6.0 ± 2.8	6.2 ± 2.8	0.493 ^d^	6.1 ± 2.8	6.2 ± 2.9	0.047	
**Extra-scleral extension**, n (%) *	9 (3.3)	103 (5.2)	0.230 ^b^	9 (3.3)	13 (4.8)	0.075	
**TNM classification**, n (%) *			0.981 ^b^			0.059	
T1	21 (7.7)	147 (7.4)		21 (7.8)	19 (7.0)		
T2	84 (30.9)	591 (29.9)		82 (30.4)	78 (28.9)		
T3	79 (29.0)	589 (29.8)		79 (29.3)	78 (28.9)		
T4	88 (32.4)	652 (32.9)		88 (32.6)	95 (35.2)		
**Other tumor characteristics**, n (%)							
Disruption of Bruch’s membrane	43 (15.8)	413 (20.9)	0.062 ^b^	43 (15.9)	61 (22.6)		0.02 ^e^
Disruption of the retina	18 (6.6)	240 (12.1)	**0.010** ^b^	18 (6.7)	25 (9.3)		0.26 ^e^
Pigment dispersion	28 (10.3)	238 (12.0)	0.466 ^b^	28 (10.4)	20 (7.4)		0.21 ^e^
Hemorrhage	22 (8.1)	311 (15.7)	0.158 ^b^	22 (8.2)	34 (12.6)		0.10 ^e^
Invasion of the optic disc	28 (10.3)	164 (8.3)	0.320 ^b^	28 (10.4)	23 (8.5)		0.46 ^e^
Invasion of the macula	48 (17.6)	253 (12.8)	**0.034** ^b^	48 (17.8)	41 (15.2)		0.41 ^e^
**Categorical distance between tumor and optic disc**, n (%)			0.069 ^b^				
Touching the optic disc	58 (21.3)	338 (17.1)		58 (21.5)	46 (17.0)		
Not touching but ≤3.5 mm from optic disc	60 (22.1)	548 (27.7)		60 (22.2)	75 (27.8)		
>3.5 mm from optic disc	154 (56.6)	1093 (55.2)		152 (56.3)	149 (55.2)		
**Categorical distance between tumor and fovea**, n (%)			0.302 ^b^				
Subfoveal	72 (26.5)	442 (22.3)		72 (26.7)	69 (25.6)		
Not subfoveal but ≤3.5 mm from fovea	70 (25.7)	553 (27.9)		70 (25.9)	60 (22.1)		
>3.5 mm from fovea	130 (47.8)	984 (49.7)		128 (47.4)	141 (52.2)		

* Variables used for matching are marked with an asterisk. Country of residence: bordering country: Italy, France, Germany, or Liechtenstein; other: any country not sharing a border with Switzerland. LTD: largest tumor diameter; STD: smallest tumor diameter; BCVA: best-corrected visual acuity; SMD: standardized mean difference (only reported for matched variables). The *p*-values are for ^a^ Mann–Whitney test, ^b^ chi-squared test with Yate’s correction, ^c^ Fisher exact test, ^d^ Student’s *t*-test, and ^e^ McNemar’s test.

**Table 2 cancers-15-04652-t002:** Clinical outcomes in the AYA and matched elder adult patient groups.

	AYAs	Elder Adults	*p* Value
**Patients, n**	**270**	**270**	
**Ocular Follow-up**			
Median ocular follow-up in years (range)	4.9 (0–20.8)	3.7 (0–16.0)	
Mean ocular follow-up in years (SD)	5.7 (4.7)	4.5 (4.0)	**0.0016**
Patients with ≥6 months of ocular follow-up, n (%)	235 (87.0)	217 (80.4)	**0.034**
Patients with ≥5 years of ocular follow-up, n (%)	132 (48.9)	113 (41.9)	0.084
Patients with useful vision (BCVA ≥ 0.1 (20/200) 5 years after treatment, n (%)	74 (56.0)	59 (52.2)	0.724
Patients with good vision (BCVA ≥ 0.6 (20/33) 5 years after treatment, n (%)	47 (35.6)	36 (31.9)	0.102
Ocular complications at last visit, n			
Rubeosis	34	38	
Pupillary seclusion	15	18	
Cataract	96	166	
Hemorrhage	59	61	
Actinic maculopathy	79	80	0.358
Actinic neuropathy	36	41	0.296
Local recurrence, n (%)	9 (3.3)	5 (1.6)	0.960
Enucleation, n (%)	15 (5.6)	9 (3.3)	
**Systemic Follow-up**			
Median follow-up to last info in years (range)	5.1 (0–20.8)	4.0 (0–16.0)	
Mean follow-up to last info in years (SD)	5.9 (4.6)	4.6 (4.0)	**0.0006**
Metastasis discovery, n (%)	33 (12.2)	19 (7.0)	0.214
Median time to metastasis discovery in years (range)	2.5 (0–9.3)	2.5 (0.6–11.8)	
Mean time to metastasis discovery in years (SD)	3.5 (2.5)	3.7 (2.9)	
Alive at 5 years, n (%)	138 (51.1)	119 (44.1)	
Dead at 5 years, n (%)	8 (3.0)	6 (2.2)	
Lost to follow-up at 5 years, n (%)	124 (45.9)	145 (53.7)	0.058
5-year OS in % (95% CI) (according to cumulative survival)	95.5% (92.4–98.6%)	96.6% (94.0–99.3%)	
10-year OS in % (95% CI) (according to cumulative survival)	94.6% (91.1–98.2%)	91.4% (86.0–97.2%)	
Confirmed death, n (%)	10 (3.7)	10 (3.7)	0.82

BCVA: best-corrected visual acuity. OS: overall survival. The *p*-values are for McNemar’s test except for the follow-up times (paired *t*-test) and actinic maculopathy, actinic neuropathy, local recurrence, and metastasis discovery (competing risk regression: Fine–Gray model for clustered data).

**Table 3 cancers-15-04652-t003:** Recent publications on young (pediatric and AYA) UM patients.

First Author [Ref] Publication Year	Age Category in Years	Patients, n	Mean Age in Years (Range)	Mean Follow-Up in Years (Range)	Metastasis Incidence at 10 Years	Overall/Relative Survival at		Comments
						5 Years	10 Years	
Singh AD [27] 2000	≤20	63	16 (3–20)	4.3 ^a^ (1–25.6)	23% ^b^	95%	89%	Overall survival at 15 years: 77%
Pogrzebielski A [4] 2006	≤20	11	17.9 (12–20)	5.1 (2.5–10.4)	0%	100%	n/a	
Vavvas D [28] 2010	≤20	17	18.1 (11–20)	16.4 (4.7–25)	0% ^c^	100% ^c^	100% ^c^	Controls >20 matched for largest tumor diameter, pigmentation of tumor, presence of symptoms, eye color, and tumor location.
	>20	51	57 ^a^ (26–81)	16.7 ^a^ (3–25)	n/a	98% ^b,c^	92% ^b,c^	
Shields CL [29] 2013	≤20	122	15 (3–20)	n/a	0.088	n/a	n/a	Metastasis incidence at 20 years: 20.2%
Kaliki S [26] 2013	Young ≤ 20	122	12 (3–20)	5.25 (0–23)	8% (18% ^c^)	n/a	n/a	Controls >20 matched for gender, tumor and anterior margin location, largest basal diameter, thickness, and extraocular extension
	Mid-adults 21–60	122	n/a	3 (0–24)	26% (21% ^c^)	n/a	n/a	
	Older adults > 60	122	n/a	2.8 (0–19)	24% (33% ^c^)	n/a	n/a	
Petrovic A [3] 2014	Young ≤ 20	43	17.3 (9–20)	12.9 (0.5–28)	11%	93%	93%	Controls >20 matched for anterior margin location, largest basal diameter, and thickness No metastasis before age 16 (n = 14)
	Adults > 20	129	50.4 (29–81)	6.6 (0.3–23.4)	34%	77%	65%	
Al-Jamal RT [30] 2014	<25	18	19.3 (13–24)	13.4 ^b^ (0.6–37)	24% ^b,c^	76% ^c^	76% ^c^	Overall survival at 15 years: 68% (i.e.: <18 y: 100%; 18–20 y: 80%; 21–24 y: 58%)
Al-Jamal RT [5] 2016	<18	114	15.1 ^a^ (2–17)	6.6 ^a^ (0–41)	n/a	97% ^c^	91% ^c^	No metastases before age 10 (n = 15)
	18–24	185	21.9 ^a^ (18–24)	5.1 ^a^ (0–37)	n/a	89% ^c^	78% ^c^	
Fry M [8] 2019	≤20	18	16.6 (4–20)	8.5 (0.2–21.5)	56% ^c^	69% ^c^	52% ^c^	
Present study 2023	AYA 15–39	270	32.5 (16–39)	5.9 (0–20.8)	19.70%	96%	95%	First study on AYAs with UM Propensity score matching
	≥40	270	60.5 (40–91)	4.6 (0–16)	12.70%	97%	91%	

n/a, not available; ^a^ median (instead of mean); ^b^ estimate from published Kaplan–Meier curve; ^c^ after exclusion of iris melanoma patients. Underlined values are for relative survival, while values without underline are for overall survival.

## Data Availability

Data of this study are available via the corresponding author upon reasonable request.

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
