# Peer review of "Clinical Outcomes in AYAs (Adolescents and Young Adults) Treated with Proton Therapy for Uveal Melanoma: A Comparative Matching Study with Elder Adults"

_cancers, 2023, doi:10.3390/cancers15184652_

Round 1

Reviewer 1 Report

this is an interesting paper

i would suggest that iris tumours are considered separately (or excluded) as they have very different metastatic behaviour to choroidal/ciliary body lesions 

table 1 and 2 can be combined and explained in the text; they are essentially duplicates with minor modifications

Author Response

We thank the Reviewers for their time to read our manuscript and their valuable input.

Below, please find the answer to each comment, point-by-point.

Reviewer 1 - I would suggest that iris tumours are considered separately (or excluded) as they have very different metastatic behaviour to choroidal/ciliary body lesions

Before performing our analysis, we have pondered this question and decided to include all uveal melanomas, including iris, for the following reasons:

  • The numbers of iris melanoma patients– as you noted as well – were too small to do a meaningful survival analysis. On the other hand, by excluding them, this article would have been only about choroidal and ciliary body melanomas, leaving the comparison between AYAs and adults regarding iris melanoma outcomes simply unaddressed.
  • By matching an Elder patient to each AYA, using, among others, two variables for tumor location and two for size, we tried to make the groups comparable. Indeed, we have a similar number of iris tumors in both matched groups. On the other hand, the numbers are not exactly the same as the propensity score matching found no appropriate match for 2 AYAs – with iris melanoma – and matched some iris melanomas to non-iris melanomas when taking into account all variables used for matching.
  • A recent paper by the group of Gustav Stålhammar (Sabazade et al, BMC Cancer (2021) 21:1270) questions the difference in long-term prognosis between iris and choroidal melanomas when adjusting for tumor thickness and diameter. It might be interesting to explore whether we could find similar results within our patients, pooling all ages and extending our analysis further into the past, to obtain a sufficient number of iris melanoma patients. However, such an analysis would be part of a different project.

Reviewer 1 - table 1 and 2 can be combined and explained in the text; they are essentially duplicates with minor modifications

We found this an excellent idea and have combined Tables 1 and 2 as suggested. Please note that, as the statistical tests differ for unmatched and matched data, we now use five different tests in one table to produce the various p-values. For reasons of clarity, we now indicate the test used for each p-value in a footnote. Please also note that we took the opportunity to correct a glitch in reporting the Year-of-treatment: we now report the median +IQR for both, before and after matching (instead of mean+SD).

Reviewer 2 Report

Pica et al. conducted a matching study comparing AYAs and elderly adults with regards to the clinical outcomes of proton therapy for uveal melanoma. The manuscript demonstrates a clear and professional use of the English language, making it easy to follow. Additionally, the authors effectively present their data in the results section through well-organized tables and easily understandable figures. They have also crafted a comprehensive discussion section that addresses various factors influencing the study outcomes, including demographic and patient characteristics, as well as tumor characteristics. Furthermore, the authors are commendable for acknowledging the limitations of their research in the discussion section. However, I would like to offer a few additional comments:

(1) Table 4: It appears that some of the text in the "comments" column was cut off. Please fix it.

(2) Introduction section: It would be better to include a brief introduction that provides background information about proton therapy. 

(3) Discussion section: To enhance the impact of their study, the authors could address the clinical and research implications of their findings. Discussing how this study might contribute to effective strategies for public policy and personalized healthcare could underscore its significance and broader relevance.

Author Response

We thank the Reviewers for their time to read our manuscript and their valuable input. Below, please find the answer to each comment, point-by-point.

Reviewer 2 - Table 4: It appears that some of the text in the "comments" column was cut off. Please fix it.

We are very grateful to reviewer 2 to have seen this formatting error and have fixed the table (now: Table 3) accordingly.

Reviewer 2 - Introduction section: It would be better to include a brief introduction that provides background information about proton therapy.

We thank Reviewer 2 for this valuable feedback. In order to better explain PT to readers not familiar with it, we have added the following sentences:

Radiation therapy (RT) represents the standard care for the vast majority ofUM.. Proton therapy (PT) is the most widely used form of RT because of its intrinsic physical properties, whereby the deposition of most energy occurs at the target tissue with a reduction of the dose proximal and distal to the tumor. PT is associated with high local tumor control rates and has been used for several decades in our center as the preferred RT modality of  for UM . 

Reviewer 2 - Discussion section: To enhance the impact of their study, the authors could address the clinical and research implications of their findings. Discussing how this study might contribute to effective strategies for public policy and personalized healthcare could underscore its significance and broader relevance

As we see it, implications of this study, apart from being the first to focus on UM outcomes in AYAs, could be linked to the confirmation of an apparent ‘threshold’ age for UM metastasis, rather than a metastatic risk gradually increasing with age. We do not know why, as age-specific molecular and genetic features are poorly understood. Ideally, all patients should undergo mutation profiling as part of their diagnostic work-up to identify tumor biology and prognostic biomarkers for metastatic disease. As this currently requires an intraocular tumor biopsy, such an approach is limited in the presence of small-sized tumors as well as by the risks of sub-retinal or vitreous hemorrhage and tumor dissemination. More promising would be the implementation of less-invasive strategies to determine the metastatic potential of UM patients in the absence of a tumor biopsy.